materials science

magnetic field, brush plating, preferred orientation, Ni/nano-Al$_2$O$_3$ coatings

**Author for correspondence:**
Cui Xiufang
e-mail: cuixf721@163.com

# Synthesis of Ni/nano-Al$_2$O$_3$ coatings by brush plating with magnetic fields

Tan Na[1], Liu Jin[1], Lou Liyan[1], Jin Guo[2], Cui Xiufang[2] and Wang Yang[2]

[1]National-Local Joint Engineering Laboratory of Intelligent Manufacturing Oriented Automobile Die and Mould, Tianjin University of Technology and Education, Tianjin 300222, People's Republic of China
[2]Institute of Surface/Interface Science and Technology, Key Laboratory of Superlight Material and Surface Technology of Ministry of Education, Harbin Engineering University, Harbin 150001, People's Republic of China

In the process of electrodeposition, the magnetic field would generate the magneto hydrodynamic (MHD) effect, which affects the flow and mass transfer of hydrogen evolution. Thus, the performance of electrodeposition coatings will be affected. In this work, Ni/nano-Al$_2$O$_3$ coatings prepared by brush plating with magnetic fields were studied. The results show that the magnetic field indeed influences the morphology, texture and wears resistance. The morphology of Ni/Al$_2$O$_3$ coating is smooth and uniform; the (200) plane of Ni/Al$_2$O$_3$ coating is preferentially oriented in the same direction of a magnetic field; the wear resistance of Ni/Al$_2$O$_3$ composite coatings increases due to the uniform distribution of Al$_2$O$_3$ particles; and the wear mechanism of coating belongs to adhesive wear. The optimal intensity of magnetic field for Ni/Al$_2$O$_3$ composite coatings to obtain a good performance is 0.1 T.

## 1. Introduction

Brush plating is a convenient technique to control the surface morphology and crystal orientation of thin metal films [1–3]. Ni composite plating has been studied extensively, such as Ni/SiC, Ni/WC and Ni/SiO$_2$. Via the process of brush plating, the surface of coating becomes smoother, and the micro-hardness, wear resistance and corrosion resistance are improved considerably [1,4–5]. A nickel sulfate bath containing SiC particles is used to enhance the hardness and wear resistance [6]. The micro-hardness of the Ni-NCZ (i.e. nickel-nickel coated ZrO$_2$) composite coating is greater than that of Ni-ZrO$_2$ due to a larger amount of co-electrodeposited particles of smaller crystallite size. The corrosion resistance of Ni-NCZ is higher than that of Ni-ZrO$_2$ [7–9].

**Figure 1.** Schematic representation of magnetic brush plating device.

The magnetic field, as a special kind of physical field, has been widely applied. And Zhang *et al.* reported that the difference in the direction of the magnetic field could affect the electrochemical deposition rate and morphology [10]. At a given current intensity, the deposition rate will increase due to magneto hydrodynamic (MHD) effect. Meanwhile, Meck found that the magnetic field inhibits the hydrogen evolution reaction [11]. The MHD effect makes the appearance of hydrogen holes decrease in the coating [12]. The Lorentz force in the magnetic and the electric field, caused by the interaction of the magnetic field force hydrodynamics (MHD) effects with the electric brush during the deposition of metal particles, has a significant effect. Koza *et al.* and Zhou & Zhong found the MHD effects mainly influence the structure and properties of the coating in two aspects, the first one is the flow and mass transfer and the other is the diffusion of Ni ions [12–14]. MHD effects on the bath particles cause a magnetic stirring, which affects the micro-structure of coating and the preferred orientation of coating [15–17].

In our work, Ni/nano-$Al_2O_3$ coatings were prepared by brush plating with magnetic fields. And the effects of magnetic fields (0 T ∼ 0.4 T magnetic flux density) on wear resistance, morphology and crystal orientation are mainly investigated.

## 2. Experimental

The electroplating of Ni/$Al_2O_3$ composite coatings was carried out by DSD-100-S DC brushing plating with a special current source in the magnetic field at 0.1, 0.2, 0.3 and 0.4 T. The magnetic field was generated by a magnet generator (PEM-40), as shown in figure 1. The current intensity is controlled to change the magnetic field strength. Prior to brushing plating, the surface of low-carbon steel was polished with 80, 320, 1000 and 2000 # silicon carbide emery paper. The chemical composition of the low-carbon steel is given in table 1.

Electric brush plating was made at a working voltage of 10–12 V. Pot leads which dip the brushing plating solution were used as an anode. The cathode was low-carbon steel (30 × 30 mm) mounted parallel to the anode plane. All experiments were performed at a temperature of 20°C. Each experiment was carried out with a fresh solution in the magnetic fields. The chemical composition of the electrode position bath is given in table 2. The size of $Al_2O_3$ particles was 5 ∼ 10 nm. The molecular weight was 101.96 g mol$^{-1}$. The content of $Al_2O_3$ particles in the deposition bath was 20 wt%.

The surface morphological examination was carried out by Quant 200 scanning electron microscopy (SEM). At the same time, the component analysis was conducted by EDAX genesis X-ray energy disperse spectroscopy (EDS). The micro-structure and the SAED pattern were examined by Philips transmission electron microscope (TEM). To obtain the porosities of the coatings statistically, they were calculated with an image analysis software (ImageJ2x) via 10 randomly selected SEM photos collected from the coatings

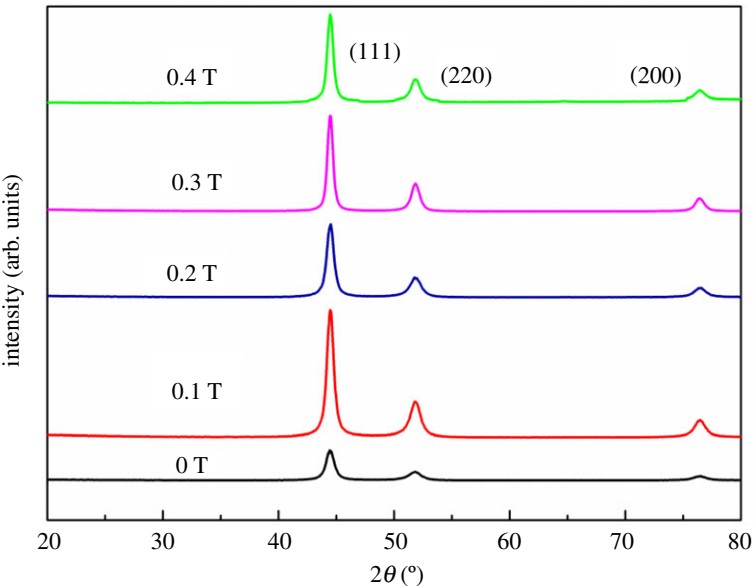

**Figure 2.** X-ray diffraction patterns of Ni/Al$_2$O$_3$ brush plating electrodeposited in the magnetic fields at 0, 0.1, 0.2, 0.3 and 0.4 T.

**Table 1.** The chemical composition of the low-carbon steel.

| matrix | C | Mn | Si | Cr | Ni | Mo | Nb | Cu | S | p |
|---|---|---|---|---|---|---|---|---|---|---|
| FV520B (wt%) | 0.02 ∼ 0.07 | 0.3∼1.0 | 0.15 ∼ 0.70 | 13.0 ∼ 14.5 | 5.0 ∼ 6.0 | 1.3 ∼ 1.8 | 0.25 ∼ 0.45 | 1.3 ∼ 1.8 | ≤0.025 | ≤0.03 |

**Table 2.** The chemical composition of the electrodepositing bath.

| NiSO$_4$ 7H$_2$O | NH$_3$ H$_2$O | (NH$_4$)$_3$C$_6$H$_5$O$_7$ | CH$_3$COONH | (COONH$_4$)$_2$H$_2$O | Al$_2$O$_3$ bath |
|---|---|---|---|---|---|
| g l$^{-1}$ | (25%) ml l$^{-1}$ | g l$^{-1}$ | g l$^{-1}$ | g l$^{-1}$ | (20%)ml l$^{-1}$ |
| 250 ∼ 255 | 100 ∼ 110 | 55 ∼ 56 | 22 ∼ 23 | 0.1 ∼ 0.2 | 50 |

with 500 × magnification. The phase structure of coatings was measured by an automatic X-ray diffractometer made by Philips Company, Cu target as ray sources. The parameters were as follows: Kα λ is 1.54056 Å, voltage 45 kV, current 100 mA, scanning speed 4° min$^{-1}$, scanning angular resolution 0.05° and scanning scope 20° ∼ 80°.

The hardness of the coating was measured by CETR-APex nano-indentation tester. The parameters were as follows: the maximum applied load was 15 mN; loading time was 60 s. At least four points were made for each specimen. The friction and wear tests of composites at room temperature were performed by CETR-3 friction testing machine with steel ball (GCr15) of Φ 4 mm at a friction distance of 5 mm, a normal load of 10 N, a test time of 15 min and a speed of 5 Hz under reciprocating sliding mode. After the friction and wear tests, the morphology of wear track was performed by SEM.

# 3. Results and discussion

## 3.1. Micro-structure characterization

Figure 2 shows the diffraction patterns in the magnetic fields at 0, 0.1, 0.2, 0.3 and 0.4 T. It can be seen that with the increase of magnetic fields intensity, the diffraction intensity of (111), (220) and (200) plane of Ni

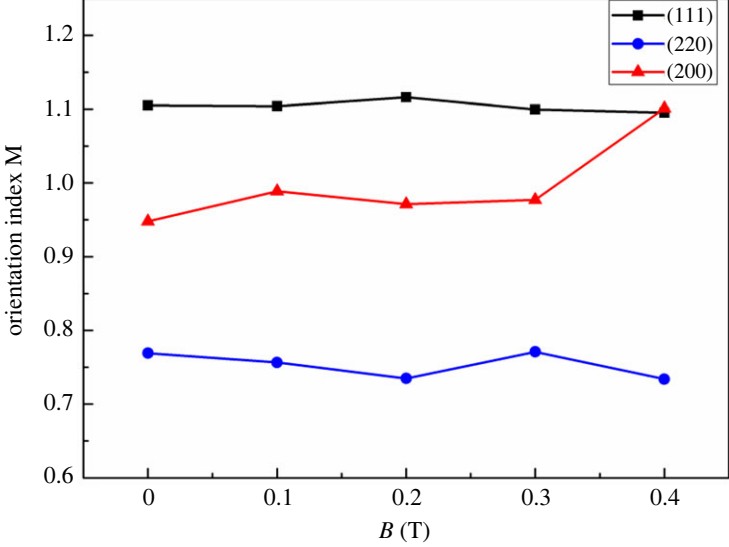

**Figure 3.** Dependence of orientation index M on magnetic field intensity for the Ni/Al$_2$O$_3$ brush plating film in the magnetic field.

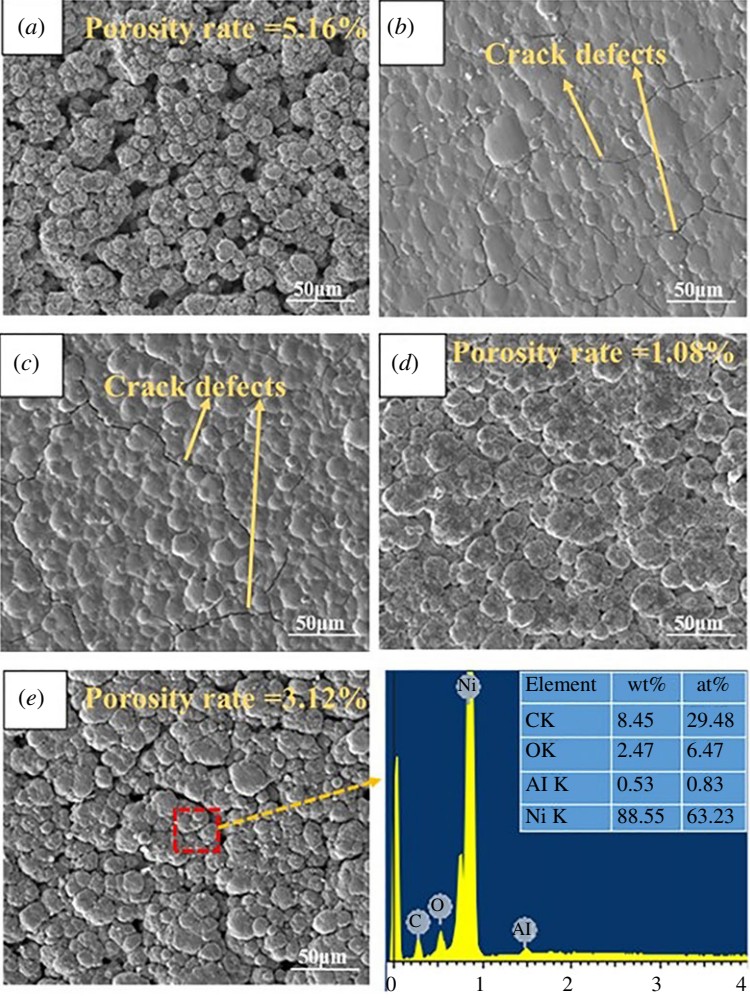

| Element | wt% | at% |
|---|---|---|
| CK | 8.45 | 29.48 |
| OK | 2.47 | 6.47 |
| Al K | 0.53 | 0.83 |
| Ni K | 88.55 | 63.23 |

**Figure 4.** Morphology of coatings under different magnetic field conditions: (*a*) 0 T Ni/Al$_2$O$_3$composite coating, (*b*) 0.1 T Ni/Al$_2$O$_3$ composite coating, (*c*) 0.2 T Ni/Al$_2$O$_3$ composite coating, (*d*) 0.3 T Ni/Al$_2$O$_3$ composite coating and (*e*) 0.4 T Ni/Al$_2$O$_3$ composite coating and EDS analysis result.

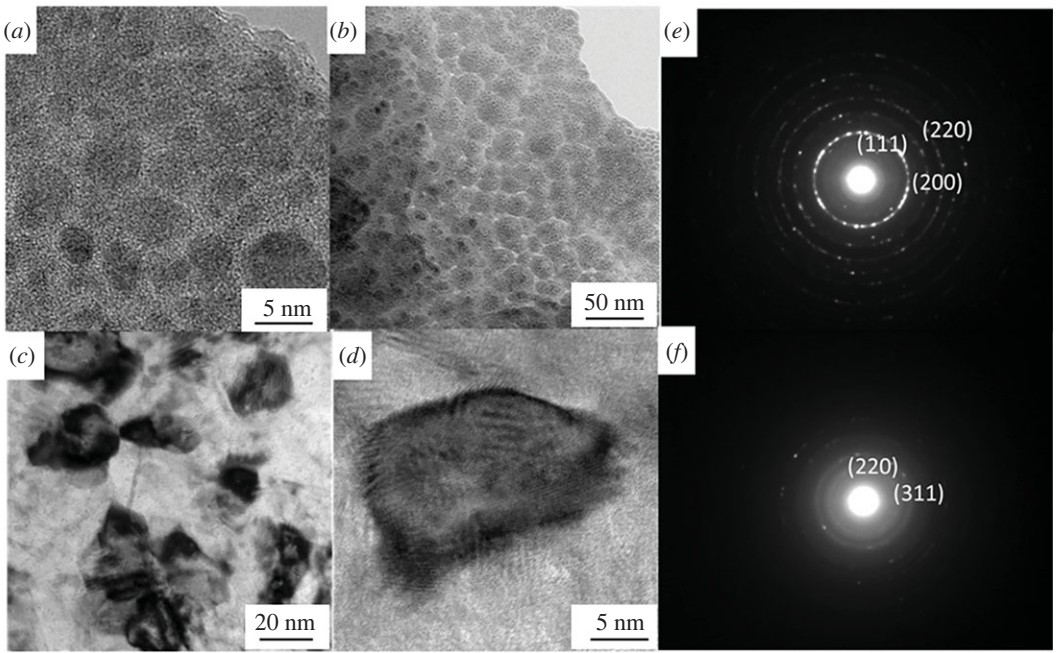

**Figure 5.** HRTEM image FETEM micrograph of brush plating. (*a*), (*b*) 0.4 T magnetic field $Al_2O_3$ brush plating coating. (*c*), (*d*) 0 T $Al_2O_3$ brush plating coating. (*e*) The SAED pattern of image of Ni in the brush plating coating. (*f*) The SAED pattern of image of $Al_2O_3$ in the brush plating coating.

become stronger. It also indicates that (111), (220) and (200) plane are the preferred orientation under the action of the magnetic field. The magnetization energy in different directions is of no balance as known to all, and the crystal tends to grow toward a stable direction. For nickel ion, it grows along the (111), (200), (220) plane in matrix surface. At the same time, the diffraction peak of $Al_2O_3$ cannot be found which may be attributed to that the nano-$Al_2O_3$ particles were enwrapped by Ni ions.

To evaluate preferred orientation paralleled to a substrate plane preferably, an orientation index M was calculated as follows [18]:

$$M(hkl) = \frac{I(hkl)/\sum I(h'k'l')}{I_0(hkl)\sum I_0(h'k'l')},$$

(3.1)

where $I(hkl)$ was X-ray diffraction (XRD) intensity in experimental data, $I_o(hkl)$ was the intensity in JCPDS cards and $M(hkl)$ was the calculated orientation index: $(h'k'l')$ in the present case was the sum of intensities of three independent peaks: (111), (200) and (220).

In the process of brush plating with the magnetic field, metal ions can be affected by magnetic field effects, electric field effects and other force fields such as a force exerted by the plating pen. Figure 3 shows the dependence of M on the magnetic field intensity. For (111) plane, the curve of M is smooth and steady basically. And for (200) and (220) planes, the M value changes clearly when magnetic field intensity increases. The (200) orientation parameter M of brush plating in the magnetic fields is larger than that of 0 T, which shows that the magnetic field promotes the (200) orientation. While when magnetic field intensity increases, the (220) orientation parameter M of brush plating in the magnetic fields falls. From the results of XRD, we can conclude that (220) plane is also the preferred orientation. In the consideration of the results in figure 2, it indicates that the crystal orientation parallel to the substrate plane depended on electric field effects and other force fields rather than magnetic field effects [19].

The surface morphology of the Ni/$Al_2O_3$ composite coatings of 0, 0.1, 0.2, 0.3 and 0.4 T is shown in figure 4. It is clear Ni, Al, O and carbon were the primary elements according to the EDS data. With further increase of magnetic field intensity, the surface morphology of Ni/$Al_2O_3$ composite coatings becomes smooth and roundish; there is no evidence of nodular agglomeration of the particles. The pore defects of coatings decrease for 0.1 and 0.2 T compared to the coating obtained without a magnetic field, while for the 0.4 T the pore defects increase compared to the coatings obtained at 0.1 and 0.2 T. Namely, 0.1 and 0.2 T are most effective magnetic flux densities to reduce the hydrogen evolution in brush plating process [20].

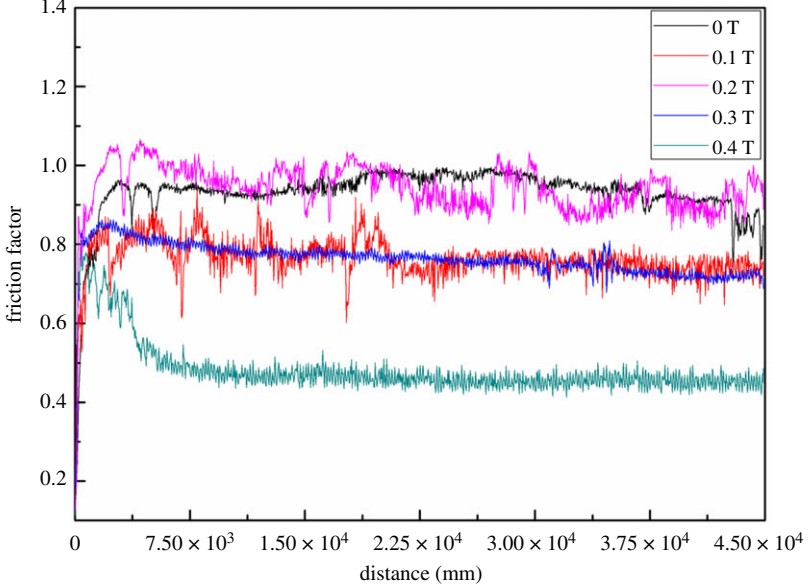

**Figure 6.** Friction factor of coatings with different magnetic field intensity.

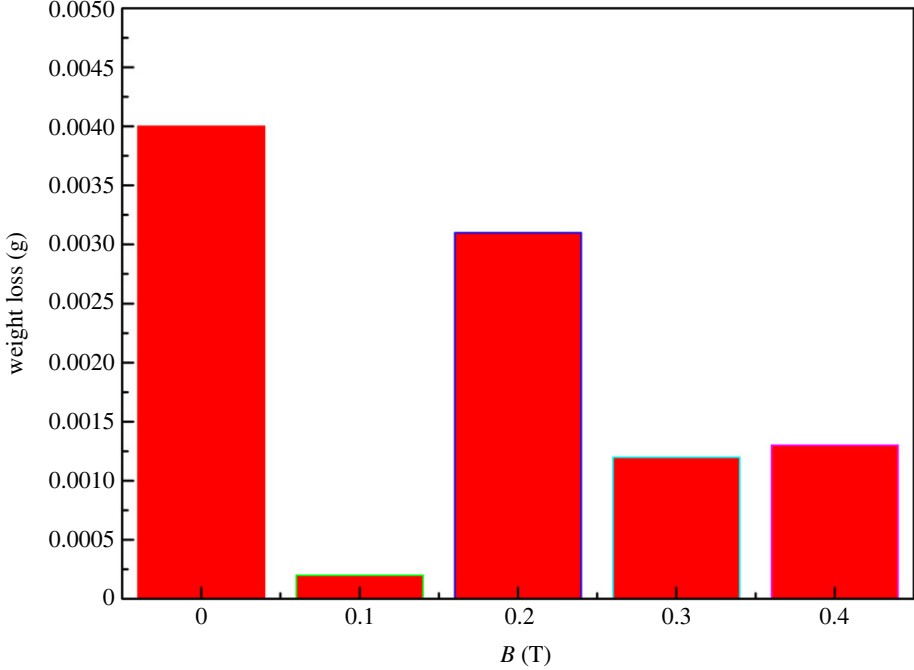

**Figure 7.** Weight loss of Ni/Al$_2$O$_3$ coating under the following conditions: 0, 0.1, 0.2, 0.3 and 0.4 T.

At the same time, the coatings obtained at 0.1 and 0.2 T seem to present cracks defects on the surface, which may attribute to the addition of the magnetic field. Microscopic stress is resulted from the magnetic field in the internal coating which can produce local micro-cracks [21]. The application of an external magnetic field on brush plating coating leads to an enhancement in the compactness of Ni grains which grow with more regular sizes and geometrical shapes [22]; the Ni/Al$_2$O$_3$ composite coating is smoother and more compact especially at 0.1 T (figure 4$b$).

Generally, crystal growth competes with nucleation in the growing layer of electrodeposition. Since the hydrogen ion is diametrical, the magnetic field can enhance the desorption of hydrogen ion on the surface. Because of MHD effect, many minute vortices appear on the surface of the coating, reducing the production of hydrogen. Desorption will make the coating surface smoother [23]. The magnetic field affects the mobility of Ni$^{2+}$ around the Al$_2$O$_3$ nanoparticles. Ni$^{2+}$ around the Al$_2$O$_3$

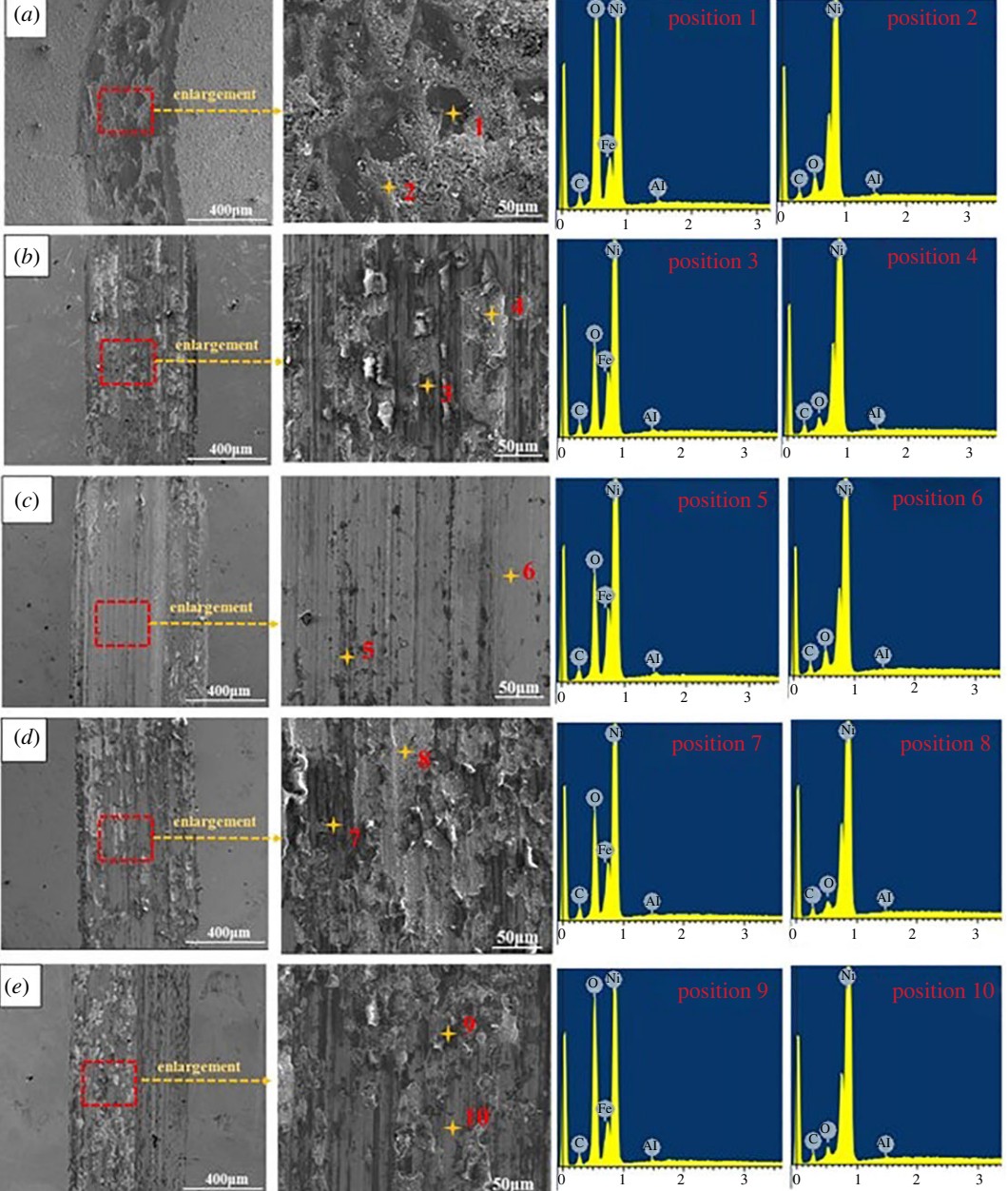

**Figure 8.** SEM surface morphology of wear orbit under different magnetic field conditions: (*a*) 0 T Ni/Al$_2$O$_3$composite coating, (*b*) 0.1 T Ni/Al$_2$O$_3$ composite coating, (*c*) 0.2 T Ni/Al$_2$O$_3$ composite coating, (*d*) 0.3 T Ni/Al$_2$O$_3$ composite coating and (*e*) 0.4 T Ni/Al$_2$O$_3$ composite coating.

nanoparticles was compensated owing to MHD effect. For Ni/Al$_2$O$_3$ composite coatings, the addition of nano-Al$_2$O$_3$ particles can increase more nucleation sites to the detriment of crystal growth and decrease the diffusion flux of Ni$^{2+}$ ions. The Ni$^{2+}$ ions and nano-Al$_2$O$_3$ particles were mainly affected by Lorentz and electric force. Therefore, the particles generate additional convection and influence deposition behaviour.

From figure 5*a* and *b*, nano-Al$_2$O$_3$ particulates are entrapped by the nickel continuously distributing in the coating, and the surface of the coating is flat and compact. The nano-Al$_2$O$_3$ particulates are spherical-like shapes. It was the magneto hydrodynamic (MHD) effect generated by magnetic field which speeds up the flow of nickel during the brushing plating and reduces hydrogen evolution [24]. Figure 5*c* and *d* shows the TEM image of nano-Al$_2$O$_3$ particulates in the 0 T coating. It indicates that the nano-Al$_2$O$_3$ particulates are irregular polygon-shaped and dispersed in the coating independently. From figure 5*e*, the (111), (220) and (200) planes are clearly seen, which agrees with the result of XRD. From the diffraction pattern (figure 5*f*), (220) and (311) crystal planes of α-Al$_2$O$_3$ are acquired by

**Table 3.** Elementary composition of different regions in the wear scars Ni/Al$_2$O$_3$ coatings (region 1–10 correspond to the corresponding region marked in figure 8).

| sample with different magnetic density | region | element composition (at%) | | | | |
|---|---|---|---|---|---|---|
| | | C | O | Ni | Al | Fe |
| 0 T | 1 | 12.34 | 55.22 | 28.38 | 0.17 | 3.9 |
| | 2 | 19.80 | 8.72 | 70.93 | 0.56 | / |
| 0.1 T | 3 | 19.82 | 30.24 | 49.49 | 0.34 | 0.10 |
| | 4 | 23.57 | 5.16 | 71.05 | 0.21 | / |
| 0.2 T | 5 | 16.65 | 34.09 | 48.62 | 0.48 | 0.16 |
| | 6 | 22.25 | 8.62 | 69.08 | 0.06 | / |
| 0.3 T | 7 | 17.53 | 35.94 | 44.74 | 0.17 | 1.62 |
| | 8 | 21.39 | 5.02 | 73.33 | 0.26 | / |
| 0.4 T | 9 | 11.99 | 48.84 | 35.84 | 0.17 | 3.15 |
| | 10 | 20.46 | 5.00 | 74.16 | 0.38 | / |

calculating identity distance. The orientation of (220) and (311) are different in the magnified edge area which is the preferred orientation in the magnetic field brushing plating. The measured value of the angle is 64.5° and the diameter of nano-Al$_2$O$_3$ is 10 ∼ 15 nm.

## 3.2. Properties analysis

The result of friction and wear experiments (figure 6) provides a comparison of dry friction coefficients. Compared with the 0 T coating, the coating in the magnetic field has a low friction coefficient. When magnetic field intensity is 0.4 T, the friction coefficient of the coating is at a minimum and steady; the friction coefficients of 0.1 and 0.3 T are similar. The 0.4 T coating has better uniformity and self-lubricity whose sliding friction coefficient is about 1/5 times of 0 T coating.

The variation in the extent of wear for the Ni/Al$_2$O$_3$ composite coatings in the magnetic fields is shown in figure 7. In contrast with brush plating coating, the wear extent of Ni/Al$_2$O$_3$ coatings decreases, which is an attribute to the magnetic field. That is to say, the MHD effect increases the particle content in the coating, and the co-deposition Al$_2$O$_3$ particles significantly increase weakening the wear behaviour. But the extent of wear for 0.1 T is minimum. As is shown in figure 4b, the dispersed Al$_2$O$_3$ of 0.1 T is more uniform than 0.2, 0.3 and 0.4 T, which may lead to better wear resistance. The results in figures 6 and 7 show that the magnetic field can effectively improve the wear resistance of the coating. The wear track and friction coefficient are investigated by the width of the track and wear debris.

Figure 8 is the Ni/Al$_2$O$_3$ coating in the magnetic field which presents SEM surface morphology of the wear orbit under 0, 0.1, 0.2, 0.3 and 0.4 T Ni/Al$_2$O$_3$ brush plating coating. The surface coating after the wear experiment presents obvious chinks. The friction pair embeds into the coating along a certain direction and circularly rolls on the surface grain coating, deforming the coating. The wear orbit of coating is shallow and the width of orbit is broad, due largely to the scattering of Al$_2$O$_3$ particles on the surface of the coating, which improves the wear resistance of brush plating coating in the magnetic field. From the SEM morphologies of the wear scars, especially from the high-magnification photographs, the adhesive wear is easy to be observed and two regions with different contrast can be obviously found, which are marked as 1–10 in figure 8a–e. The wear mechanism of Ni/Al$_2$O$_3$ coating is adhesive wear [25,26].

Table 3 displays the elementary composition of different contrast regions marked in figure 8. It can be seen that Ni, O, Al, C and Fe elements are the primary elements according to the EDS data in regions 1, 3, 5, 7 and 9. The appearance of Fe element indicates the materials transfer which is in accordance of GCr15 counterbody. while Ni, O, C and Al elements are the primary elements in regions 2, 4, 6, 8 and 10. Thus, it can be inferred that the formation of regions 1, 3, 5 and 7 is due to the tribo-oxidation reaction during the friction and wear test. The obvious increment of O content indicates that tribo-oxidation occurred

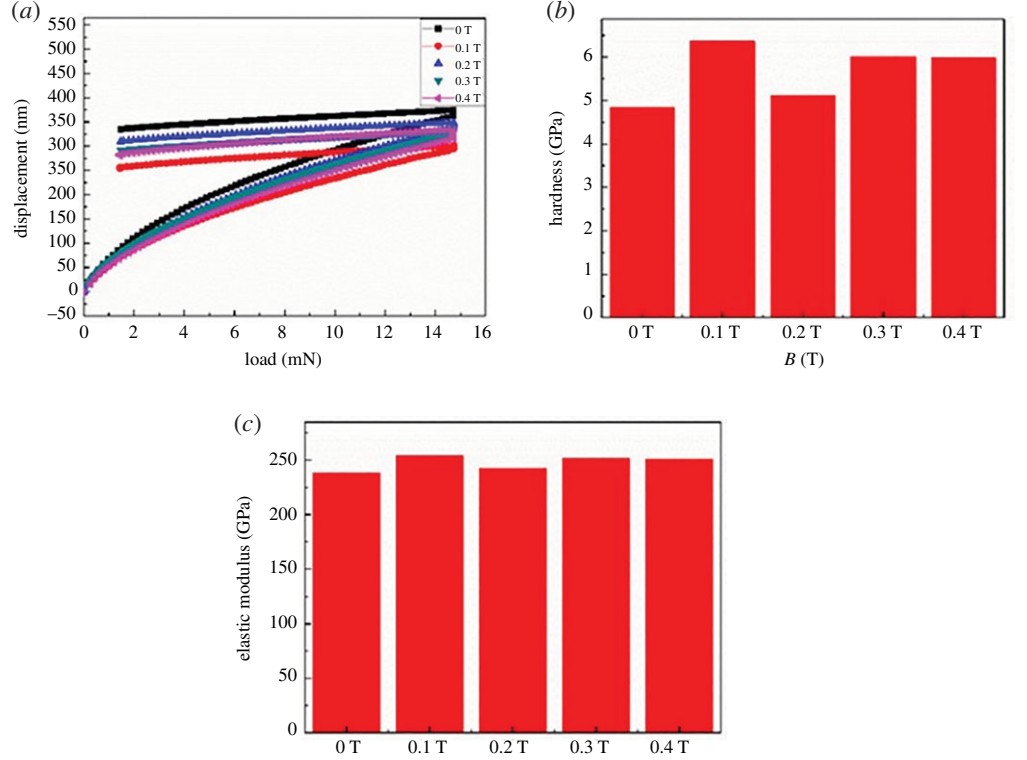

**Figure 9.** (*a*) Load-displacement curve of magnetic field Ni/Al$_2$O$_3$ brush plating, (*b*) hardness of magnetic field Ni/Al$_2$O$_3$ brush plating and (*c*) elasticity modulus of magnetic field Ni/Al$_2$O$_3$ brush plating.

during the wear test [26,27], while regions 2, 4, 6 and 8 are mainly composed of the original film which is in accordance with the EDS analysis result of figure 4*e*.

From nano-indentation test results (figure 9), the coatings in the magnetic field significantly enhance hardness compared with the Ni/Al$_2$O$_3$ brush plating without a magnetic field. From the nano-indentation load-displacement curve, when the applied load is at a maximum with the same loading rate; the indentation depth of brush plating without magnetic field is at a maximum and the indentation depth of 0.1 T brush plating is a minimum. It shows that the hardness and modulus of elasticity are at a maximum. The hardness and elastic modulus of the coating surface are mainly due to stress and dislocation distribution. The morphology of 0.1 T is smooth and the blowhole defects become less, therefore the hardness and modulus of elasticity are at a maximum.

## 4. Conclusion

The influence of magnetic field on the brush plating of Ni/Al$_2$O$_3$ is studied. The results show that the magnetic field influences the morphology, texture and wears resistance of the coating.

- (i) The morphology of Ni/Al$_2$O$_3$ coating is smooth and uniform. The magnetic field effectively promotes the co-deposition of Al$_2$O$_3$ particles with nickel to achieve a more uniform distribution of Al$_2$O$_3$ particles in metal matrix.
- (ii) The (200) plane of Ni/Al$_2$O$_3$ coating is preferentially oriented in the same direction of a magnetic field.
- (iii) The wear resistance of Ni/Al$_2$O$_3$ composite coatings in the magnetic field increase due to the uniform distribution of Al$_2$O$_3$ particles. And the wear mechanism of coating belongs to adhesive wear.
- (iv) 0.1 T may be a good magnetic flux density for Ni/Al$_2$O$_3$ composite coatings to obtain a good performance.

Data accessibility. Data are available at the Dryad Digital Repository: https://doi.org/10.5061/dryad.9cnp5hqgr [28].

Authors' contributions. L.J. and W.Y. carried out the laboratory work; L.L. and J.G. carried out the statistical analyses; C.X. participated in the design of the study. All authors gave final approval for publication.

Competing interests. We declare we have no competing interests

Funding. This work was financially supported by the Natural Science Foundation of Heilongjiang Province (no. E2018020), the Fundamental Research Funds for the Central Universities (no. HEUCFG201838) and Tianjin Natural Science Foundation (19JCQNJC03800).

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
