## [Peer Review File · Royal Society Open Science]

Review History

RSOS-202089.R0 (Original submission)

Review form: Reviewer 1

Is the manuscript scientifically sound in its present form?

Yes

Are the interpretations and conclusions justified by the results?

Yes

Is the language acceptable?

Yes

Do you have any ethical concerns with this paper?

No

Have you any concerns about statistical analyses in this paper?

No

Recommendation?

Accept with minor revision (please list in comments)

Comments to the Author(s)

In this manuscript, the authors have synthesized Ni/nano-Al₂O₃ coatings by brush plating under magnetic assistance. Their morphology, phase composition and wear behavior were and characterized. The interesting work is well described and the paper is well organized.

Nevertheless, the following issues should be properly addressed before it can be accepted. .

1. The operating parameters of friction test such as load, friction time and the speed should be provided in the Experimental.
2. The moving direction of brush plating pen should be informed.
3. The fourth conclusion 0.1T "may be a good magnetic field" is not suitable. or magnetic intensity ? ?
4. The table caption should be located at the top of the tables.
5. Scale bar including the font in Fig.4, 7, 8 should be enlarged and presented clearly.
6. In the text, the insert position of the tables and pictures needs to be marked
7. The section 3 "Results and discussion" is suggested to write in 3.1, 3.2 structure.

Review form: Reviewer 2**Is the manuscript scientifically sound in its present form?**

No

Are the interpretations and conclusions justified by the results?

No

Is the language acceptable?

Yes

Do you have any ethical concerns with this paper?

No

Have you any concerns about statistical analyses in this paper?

No

Recommendation?

Major revision is needed (please make suggestions in comments)

Comments to the Author(s)

Dear Authors,

The paper is interesting, however a major corrections are necessary.

1. Provide the name of the nanoindenter used in the experiment.
2. In the Experimental part, there is no description of the tribological test. Provide the name and parameters used for the test. What was a counterbody? What was the load? What was the distance of the test?
3. The authors stated that the porosity decreased with and increase of the magnetic field intensity. How was the porosity calculated?
4. All the provided micrographs of the wear tracks should be done the same magnification.
5. Provide higher magnification of worn surface in order to reveal the wear mechanisms.

6. What was the chemical/phase composition of the wear debris? Have the authors observed any film formation or material transfer?
7. All the obtained results should be discussed with up to date literature.

Decision letter (RSOS-202089.R0)

Dear Mrs Tan:

Title: Synthesis of Ni/nano-Al₂O₃ coatings by brush plating with magnetic fields
Manuscript ID: RSOS-202089

The editor assigned to your manuscript has now received comments from reviewers. We would like you to revise your paper in accordance with the referee and Subject Editor suggestions which can be found below (not including confidential reports to the Editor). Please note this decision does not guarantee eventual acceptance.

Please submit your revised paper before 05-Feb-2021. Please note that the revision deadline will expire at 00.00am on this date. If we do not hear from you within this time then it will be assumed that the paper has been withdrawn. In exceptional circumstances, extensions may be possible if agreed with the Editorial Office in advance. We do not allow multiple rounds of revision so we urge you to make every effort to fully address all of the comments at this stage. If deemed necessary by the Editors, your manuscript will be sent back to one or more of the original reviewers for assessment. If the original reviewers are not available we may invite new reviewers.

On behalf of the Subject Editor Professor Anthony Stace and the Associate Editor Professor Chaohua Cui.

RSC Associate Editor: 1
 Comments to the Author:
 (There are no comments.)

RSC Associate Editor: 2
 Comments to the Author:
 (There are no comments.)

Reviewers' Comments to Author:
 Reviewer: 1

Comments to the Author(s)

In this manuscript, the authors have synthesized Ni/nano-Al₂O₃ coatings by brush plating under magnetic assistance. Their morphology, phase composition and wear behavior were and characterized. The interesting work is well described and the paper is well organized. Nevertheless, the following issues should be properly addressed before it can be accepted. .

1. The operating parameters of friction test such as load, friction time and the speed should be provided in the Experimental.
2. The moving direction of brush plating pen should be informed.
3. The fourth conclusion 0.1T "may be a good magnetic field" is not suitable. or magnetic intensity ? ?
4. The table caption should be located at the top of the tables.
5. Scale bar including the font in Fig.4, 7, 8 should be enlarged and presented clearly.
6. In the text, the insert position of the tables and pictures needs to be marked
7. The section 3 "Results and discussion" is suggested to write in 3.1, 3.2 structure.

Reviewer: 2

Comments to the Author(s)
 Dear Authors,

The paper is interesting, however a major corrections are necessary.

1. Provide the name of the nanoindenter used in the experiment.
2. In the Experimental part, there is no description of the tribological test. Provide the name and parameters used for the test. What was a counterbody? What was the load? What was the distance of the test?
3. The authors stated that the porosity decreased with and increase of the magnetic field intensity. How was the porosity calculated?

4. All the provided micrographs of the wear tracks should be done the same magnification.
5. Provide higher magnification of worn surface in order to reveal the wear mechanisms.
6. What was the chemical/phase composition of the wear debris? Have the authors observed any film formation or material transfer?
7. All the obtained results should be discussed with up to date literature.

Author's Response to Decision Letter for (RSOS-202089.R0)

See Appendix A.

RSOS-202089.R1 (Revision)

Review form: Reviewer 1

Is the manuscript scientifically sound in its present form?

Yes

Are the interpretations and conclusions justified by the results?

Yes

Is the language acceptable?

Yes

Do you have any ethical concerns with this paper?

Yes

Have you any concerns about statistical analyses in this paper?

Yes

Recommendation?

Accept as is

Comments to the Author(s)

Thanks to the author for the responses. All the concerns have been adequately addressed and corrected according to the comments. Now, the revised manuscript can be accepted for publication in the present form.

Review form: Reviewer 2

Is the manuscript scientifically sound in its present form?

Yes

Are the interpretations and conclusions justified by the results?

Yes

Is the language acceptable?

Yes

Do you have any ethical concerns with this paper?

No

Have you any concerns about statistical analyses in this paper?

Yes

Recommendation?

Accept with minor revision (please list in comments)

Comments to the Author(s)

The paper is very interesting and the corrections were taken into consideration by authors. However, additional aspects should be considered.

- 1) Why the friction distance is only 5mm? What was the total distance of the test?
- 2) The graph with coefficient of friction should be in the function of distance not the time.
- 3) The authors provided quantitative EDS results with a precise amount of oxygen and carbon. How was it calculated? Please explain. Especially, if the EDS system should not be used for quantitative examination of light elements such as O, N, C or B due to its construction. In this case it is much more accurate to provide only spectrum instead of quantitative evaluation.

Decision letter (RSOS-202089.R1)

Dear Mrs Tan:

Title: Synthesis of Ni/nano-Al₂O₃ coatings by brush plating with magnetic fields
Manuscript ID: RSOS-202089.R1

Thank you for submitting the above manuscript to Royal Society Open Science. On behalf of the Editors and the Royal Society of Chemistry, I am pleased to inform you that your manuscript will be accepted for publication in Royal Society Open Science subject to minor revision in accordance with the referee suggestions. Please find the reviewers' comments at the end of this email.

The reviewers and handling editors have recommended publication, but also suggest some minor revisions to your manuscript. Therefore, I invite you to respond to the comments and revise your manuscript.

Because the schedule for publication is very tight, it is a condition of publication that you submit the revised version of your manuscript before 19-Feb-2021. Please note that the revision deadline will expire at 00.00am on this date. If you do not think you will be able to meet this date please let me know immediately.

To revise your manuscript, log into <https://mc.manuscriptcentral.com/rsos> and enter your Author Centre, where you will find your manuscript title listed under "Manuscripts with Decisions". Under "Actions," click on "Create a Revision." You will be unable to make your

revisions on the originally submitted version of the manuscript. Instead, revise your manuscript and upload a new version through your Author Centre.

Kind regards,
Dr Laura Smith
Publishing Editor, Journals

On behalf of the Subject Editor Professor Anthony Stace and the Associate Editor Professor Chaohua Cui.

RSC Associate Editor:
Comments to the Author:
(There are no comments.)

RSC Subject Editor:
Comments to the Author:
(There are no comments.)

Reviewer comments to Author:
Reviewer: 1

Comments to the Author(s)
Thanks to the author for the responses. All the concerns have been adequately addressed and corrected according to the comments. Now, the revised manuscript can be accepted for publication in the present form.

Reviewer: 2

Comments to the Author(s)
The paper is very interesting and the corrections were taken into consideration by authors. However, additional aspects should be considered.
1) Why the friction distance is only 5mm? What was the total distance of the test?
2) The graph with coefficient of friction should be in the function of distance not the time.
3) The authors provided quantitative EDS results with a precise amount of oxygen and carbon. How was it calculated? Please explain. Especially, if the EDS system should not be used for quantitative examination of light elements such as O, N, C or B due to its construction. In this case it is much more accurate to provide only spectrum instead of quantitative evaluation.

Author's Response to Decision Letter for (RSOS-202089.R1)

See Appendix B.

Decision letter (RSOS-202089.R2)

Dear Mrs Tan:

Title: Synthesis of Ni/nano-Al₂O₃ coatings by brush plating with magnetic fields
Manuscript ID: RSOS-202089.R2

It is a pleasure to accept your manuscript in its current form for publication in Royal Society Open Science. The chemistry content of Royal Society Open Science is published in collaboration with the Royal Society of Chemistry.

On behalf of the Subject Editor Professor Anthony Stace and the Associate Editor Professor Chaohua Cui.

RSC Associate Editor
Comments to the Author:
(There are no comments.)

Reviewer(s)' Comments to Author:

Appendix A

Responses to comments of Reviewers

Dear editor,

Thank you for your useful comments and suggestions on our manuscript entitled “Synthesis of Ni/nano-Al₂O₃ coatings by brush plating with magnetic fields”. By now we have completed the revision about the manuscript. We have made some marks on the revised manuscript in red words. Detailed corrections are listed below point by point:

Reviewer: 1

Comments 1

The operating parameters of friction test such as load, friction time and the speed should be provided in the Experimental.

Response:

We are grateful for your comment. Friction test details are added in “Experimental” part. The friction and wear tests of composites at room temperature were performed by CETR-3 friction testing machine with steel ball (GCr15) of $\Phi 4$ mm at a friction distance of 5 mm, a normal load of 10 N, a test time of 15 min and a speed of 5Hz. After the friction and wear tests, the morphology of wear track was performed by SEM. (Page 4, Paragraph 3, Line3-6 and (Page 5, Paragraph 1, Line1)

Comments 2

The moving direction of brush plating pen should be informed.

Response:

Thanks for your comment. Moving direction of brush plating pen has been indicated in figure 1. (Figure 1 (in the revised manuscript))

Fig.1 (in the revised manuscript) Schematic representation of magnetic brush plating device

Comments 3

The fourth conclusion 0.1T "may be a good magnetic field" is not suitable. or magnetic intensity?

Response:

Thanks for your comment. We went through the literature carefully. "magnetic field" has been revised to "magnetic flux density" in conclusion part. (Page 15, Paragraph 1, Line1)

Comments 4

The table caption should be located at the top of the tables.

Response:

We are sorry for the mistake. The table caption has been located at the top of the tables as shown table 1 and table 2 in revised manuscript. (table 1 and table 2)

Comments 5

Scale bar including the font in Fig.4, 7, 8 should be enlarged and presented clearly.

Response:

We are sorry for the mistake. The scale bar in figure 4,7,8 has been enlarged and presented clearly as shown figure 4,5,8 in revised manuscript. (figure 4,5,8)

Comments 6

In the text, the insert position of the tables and pictures needs to be marked.

Response:

We are grateful for your comment. Tables and pictures has been inserted in the proper position in the revised manuscript. And the figures and the tables are marked in red words in revised manuscript.

Comments 7

The section 3 "Results and discussion" is suggested to write in 3.1, 3.2 ... structure.

Response:

We are grateful for your comment. Results and discussion is rewritten in 3.1 microstructure characterization and 3.2 properties analysis structure. And the TEM characterization is inserted in 3.1 microstructure characterization part in order to be understood easily. (section 3 "Results and discussion")

Reviewer: 2

Comments 1

Provide the name of the nanoindenter used in the experiment.

Response:

Thanks for your suggestion. The name of the nanoindenter is CETR-APex nanoindentation tester and it has been provided in Experimental part. (Page 4, Paragraph 3, Line1)

Comments 2

In the Experimental part, there is no description of the tribological test. Provide the name and parameters used for the test. What was a counterbody? What was the load? What was the distance of the test?

Response:

We are grateful for your comment. Tribological test are added in “Experimental” part. The name, parameters, counterbody, load and the distance are also provided in the revised manuscript. The friction and wear tests of composites at room temperature were performed by CETR-3 friction testing machine with steel ball (GCr15) of $\Phi 4$ mm at a friction distance of 5 mm, a normal load of 10 N, a test time of 15 min and a speed of 5Hz. After the friction and wear tests, the morphology of wear track was performed by SEM. (Page 4, Paragraph 3, Line3-6 and (Page 5, Paragraph 1, Line1)

Comments 3

The authors stated that the porosity decreased with and increase of the magnetic field intensity. How was the porosity calculated?

Response:

Thanks for your comments.

Firstly, we are sorry for that our represent is not enough rigorous. The pore defects and the crack defects can be seen in figure 4. And represent about porosity has been revised in the revised manuscript highlighted in red words.

Secondly, The pore defect can be evaluated by the porosity rate. The porosity rate can be calculated by image J2x software. And the calculation method of porosity rate has been added in the experimental part: To obtain the porosities of the coatings statistically, they were calculated with an image analysis software (ImageJ2x) via ten randomly selected SEM photos collected from the coatings with 500 \times magnification.

The porosity was determined by calculating the percentage of the single-color area in the intensity image in accordance with the ASTM Standard E2109-01. A set of images for each magnetic field intensity is calculated to illustrate the computational process as shown in the following figure.

At last, the average porosity rate after calculation of ten SEM pictures under 0T, 0.3T and 0.4T magnetic flux density have been inserted in the pictures of figure 4(a), (b) and (e). Figure 4(c) and (d) present cracks defects in the coating under 0.1T and 0.2T magnetic flux density, and the pore defects are almost non-existent, Thus the porosity rate is not inserted in the picture of 4(c) and (d). However, the crack defects is indicated in the picture of 4(c) and (d).

Thanks for your kindly suggestion again.

Comments 4

All the provided micrographs of the wear tracks should be done the same magnification.

Response:

We are sorry for the mistake. Micrographs of the wear tracks in the same magnification has been provided in figure 8. (Figure 8)

Comments 5

Provide higher magnification of worn surface in order to reveal the wear mechanisms.

Response:

Thanks a lot. Higher magnification of worn surface has been provided as shown figure 8 and the wear mechanisms has been discussed based on elements analysis (Table 3). (Figure 8)

Comments 6

What was the chemical/phase composition of the wear debris? Have the authors observed any film formation or material transfer?

Response:

Thanks for you advise.

Firstly, the wear debris are collected and the EDS analysis results is as shown in the following figure. It can be shown from the following figure that the chemical

composition of wear debris is C, O, Al, Ni and Fe elements. Appearance of Fe element indicates the materials transfer which is in accordance of GCr15 counterbody. The obvious increasement of O content indicates that tribo-oxidation occurred during the wear test compared to the EDS analysis result of fig. 4e.

Element	Wt. %	At. %
C K	10.26	29.57
O K	10.95	23.71
Al K	0.29	0.38
Fe K	0.10	0.63
Ni K	77.50	45.71

Secondly, Table 3(in the revised manuscript) displays the elementary composition of different contrast regions marked in Fig. 8. It can be seen that Ni, O, Al, C and Fe elements are the primary elements according to the EDS data in the regions 1, 3, 5, 7 and 9. Appearance of Fe element indicates the materials transfer which is in accordance of GCr15 counterbody, while Ni, O, C and Al elements are the primary elements in the regions 2, 4, 6, 8 and 10. Thus, it can be inferred that the formation of regions 1, 3, 5 and 7 is due to the tribo-oxidation reaction during the friction and wear test. The obvious increasement of O content indicates that tribo-oxidation occurred during the wear test [26, 27] (in the revised manuscript). While the regions 2, 4, 6 and 8 are mainly composed of the original coating which is accordance with EDS analysis result of fig. 4e.

Reference 26 and 27 in the revised manuscript.

[26] Cai Z , Chen R , Wang W , et al. Microstructure and tribological properties of Cu-doped VCN films: The role of Cu[J]. Applied Surface Science, 2020(510):145509.

[27] Tang L , Kang J J , He P F , et al. Effects of spraying conditions on the microstructure and properties of NiCrBSi coatings prepared by internal rotating plasma spraying[J]. Surface and Coatings Technology, 2019, 374:625-633.

Table 3(in the revised manuscript) Elementary composition of different regions in the wear scars Ni/Al₂O₃ coatings (region 1–10 correspond to the corresponding region marked in Fig. 8)

Sample with different magnetic density	region	Element composition (at. %)				
		C	O	Ni	Al	Fe
0T	1	12.34	55.22	28.38	0.17	3.9
	2	19.80	8.72	70.93	0.56	/
0.1T	3	19.82	30.24	49.49	0.34	0.10
	4	23.57	5.16	71.05	0.21	/
0.2T	5	16.65	34.09	48.62	0.48	0.16
	6	22.25	8.62	69.08	0.06	/
0.3T	7	17.53	35.94	44.74	0.17	1.62
	8	21.39	5.02	73.33	0.26	/
0.4T	9	11.99	48.84	35.84	0.17	3.15
	10	2.046	5.00	74.16	0.38	/

At last, both the chemical composition of wear debris and the EDS analysis results of wear scar from Table 3 (in the revised manuscript) indicates the materials transfer and the tribo-oxidation reaction during the friction and wear test. So we only provide the EDS analysis results of wear scar. We have learned more about understanding the wear mechanisms and behavior from your kindly review. In the future study about wear mechanisms and behavior, we will concern on the SEM and EDS analysis of wear debris. Thanks for your kindly suggestion again.

Comments 7

All the obtained results should be discussed with up to date literature.

Response:

Thanks for you advise. The results and discussion section has been discussed with up to date literature. Meanwhile, in the revised manuscript, we have also replaced some new references in recent years and we have remained some classic literatures.

(Reference section)

Appendix B

Responses to comments of Reviewers

Dear editor,

Thank you for your useful comments and suggestions on our manuscript entitled “**Synthesis of Ni/nano-Al₂O₃ coatings by brush plating with magnetic fields**”. By now we have completed the revision about the manuscript and revised the corresponding parts, which is highlighted in red words. Detailed corrections are listed below point by point:

Reviewer: 2

Comments to the Author(s)

The paper is very interesting and the corrections were taken into consideration by authors. However, additional aspects should be considered.

Comment 1:

Why the friction distance is only 5mm? What was the total distance of the test?

Response:

Thank you for your comments. The friction and wear tests of composites at room temperature were performed at a friction distance of 5 mm, a normal load of 10 N, a test time of 15 min and a speed of 5Hz under reciprocating sliding mode. The 5mm distance under reciprocating sliding mode at the time of 15min is enough to provide support for friction and wear properties due to the coatings is uniform. Distance is equal to the time (15min=900s) multiplying the speed (5Hz=5recycles/s=50mm/s), thus, the total distance of the test is 45m. And the “under reciprocating sliding mode” has been added in the revised manuscript R2. (Page 4, Paragraph 2, Line 6)

Comment 2:

The graph with coefficient of friction should be in the function of distance not the time.

Response:

Thanks a lot.

The graph with coefficient of friction which is in the function of distance has been revised in revised manuscript R2. (Figure 6)

Comment 3:

The authors provided quantitative EDS results with a precise amount of oxygen and carbon. How was it calculated? Please explain. Especially, if the EDS system should not be used for quantitative examination of light elements such as O, N, C or B due to its construction. In this case it is much more accurate to provide only spectrum instead of quantitative evaluation

Response:

Thank you for your suggestions.

Firstly, the atomic mass of each element obtained by EDS test is relative atomic mass. It's more intuitive to provide the relative atomic mass.

Secondly, the X-ray photons entering the detector create electron-hole pairs in the crystal. The electron-hole pair forms a voltage pulse signal, and the height of the voltage pulse output by the detector corresponds to the energy of the X-ray. Under the same conditions, the percentage of each element can be obtained by measuring the X-ray intensity of each element in the sample simultaneously.

Thirdly, the spectrum has been provided in revised manuscript. (Figure 4 and Figure 8)